# Leveraging Passage Embeddings for Efficient Listwise Reranking with Large Language Models

## ABSTRACT

Recent studies have demonstrated the effectiveness of using large language language models (LLMs) in passage ranking. The listwise approaches, such as RankGPT, have become new state-of-the-art in this task. However, the efficiency of RankGPT models is limited by the maximum context length and relatively high latency of LLM inference. To address these issues, in this paper, we propose PE-Rank, leveraging the single passage embedding as a good context compression for efficient listwise passage reranking. By treating each passage as a special token, we can directly input passage embeddings into LLMs, thereby reducing input length. Additionally, we introduce an inference method that dynamically constrains the decoding space to these special tokens, accelerating the decoding process. For adapting the model to reranking, we employ listwise learning to rank loss for training. Evaluation results on multiple benchmarks demonstrate that PE-Rank significantly improves efficiency in both prefilling and decoding, while maintaining competitive ranking effectiveness.

## CCS CONCEPTS

• **Information systems → Language models**.

## KEYWORDS

Reranking, Large Language Models, Efficiency

**ACM Reference Format:**
Anonymous Author(s). 2018. Leveraging Passage Embeddings for Efficient Listwise Reranking with Large Language Models. In *Proceedings of Make sure to enter the correct conference title from your rights confirmation emai (Conference acronym 'XX)*. ACM, New York, NY, USA, 10 pages. https://doi.org/XXXXXXX.XXXXXXX

**Prompt:**
The following are passages related to query #{query}.
Passage 1: #{passage 1}
...
Rank these passages based on their relevance to the query.

**Output:** [2] > [3] > [1] ...

**Prompt:**
The following are passages related to query #{query}, each with a special token representing the passage enclosed in [].
Passage 1: [<p1>]
...
Rank these passages based on their relevance to the query.

**Output:** <p2><p3><p1> ...

**Figure 1: Comparison between RankGPT (upper) and PE-Rank (lower). RankGPT takes the whole passage as input and outputs ordered numbers, while PE-Rank takes a list of special tokens as both input and output. On the right side, we show the reranking results on DL19 using different forms of inputs.**

## 1 INTRODUCTION

Passage ranking, which aims to rank each passage in a large corpus according to its relevance to the user's information need expressed in a short query, is an important task in information retrieval and natural language processing and plays a crucial role in many applications such as web search and retrieval-augmented generation. To achieve both effectiveness and efficiency, current mainstream approaches usually follow a two-stage paradigm known as "*retrieval-then-rerank*", which involves efficiently retrieving a set of candidates first, and further reranking them with a reranker to boost the effectiveness [15, 22].

In the first retrieval stage, dense retrieval models based on a bi-encoder architecture are widely used [11]. Trained on large-scale datasets of text pairs through contrastive learning, these models can encode text into a low-dimensional dense embedding and capture the relevance between query and passage using vector similarity.

In the second reranking stage, we can employ more sophisticated models for better ranking performance. A common reranking model is a supervised model based on the cross-encoder design [22]. With the emergence of large language models (LLMs), such as GPT-4 [23], a series of studies have tried to leverage LLMs' text comprehension and reasoning abilities for zero-shot reranking. Typically, there are three main prompting approaches: *pointwise* [12, 29], *pairwise* [27], and *listwise* [25, 30]. Among these methods, listwise approaches like RankGPT [30] have achieved state-of-the-art performance by directly producing a final ranking list for multiple passages, rather than merely assessing the relevance of a single passage or the relative position between two passages.

While the listwise approaches demonstrate good performance in the reranking task, they are limited by two challenges. Firstly, some LLMs are limited by context length and cannot rank multiple passages simultaneously, necessitating techniques such as a sliding window strategy to complete the ranking process [30]. Secondly, incorporating entire passages into prompts significantly increases inference costs, resulting in high latency in practice [38], which is untenable in the ranking scenario.

To tackle these issues, it is imperative to compress listwise reranking prompts. Some context compression methods have been proposed for LLMs and can be categorized into two types: compressing the context into dense memory slots [3, 6, 18] and directly editing the input contexts [10]. Nonetheless, existing methods exhibit relatively low compression rates and usually only compress a single passage, rendering them inadequate for ranking tasks.

For compressing multiple passages for reranking, we first highlight that in the "*retrieval-then-rerank*" pipeline, dense retrieval models have been trained as effective text compressors with their embedding capable of representing nearly as much information as the original text [17]. In this paper, we propose a novel and efficient listwise passage reranking method named **PE-Rank**, leveraging the single embedding of the passage as the compressed representation. Specifically, we obtain the passage embedding from a dense retrieval model and regard it as a special token of the LLM to replace the original text as input. To align the embedding space of the retrieval model and the input embedding space of the LLM, we use a projector as a bridge between the two models, which is inspired by previous work about modality alignment [13].

To enable PE-Rank to complete ranking tasks, we propose novel inference and training methods. For efficient inference, we propose a "Dynamic-Constrained Decoding" strategy that dynamically changes and constrains the decoding spaces to a set of special tokens that represent the rest of the passages to be ranked. We employ two-stage training, first training the projector for modality alignment, then training both the projector and LLM for ranking tasks using listwise learning to rank loss.

We evaluate PE-Rank on popular retrieval benchmarks TREC DL and BEIR. Experimental results demonstrate that PE-Rank achieves comparable ranking performance to uncompressed methods while notably improving inference efficiency. Notably, when reranking top 100 candidates retrieval by BM25 on DL19, NDCG@10 of PE-Rank is only reduced by less than 2% compared to the uncompressed method under the same settings while reducing the latency by a factor of 4.5.

In summary, the main contributions of this paper are as follows:

- We propose a novel efficient listwise reranking method, PE-Rank, which is the first model that leverages passage embeddings for context compression and highly efficient listwise reranking.
- We propose a two-stage training strategy that includes alignment and learning-to-rank stage to effectively train PE-Rank and a novel decoding method for efficient inference.
- We evaluate PE-Rank on multiple benchmarks and show its competitive ranking performance and significant efficiency advantages.

## 2 RELATED WORK

### 2.1 Large Language Models as Rerankers

Recently, large language models have demonstrated impressive effectiveness on many tasks. Many studies also attempt to utilize LLMs for zero-shot reranking. In general, there are three paradigms for prompting LLMs: *pointwise*, *pairwise*, and *listwise*.

The pointwise approach evaluates the relevance score on one query-passage pair at a time, including *relevance generation* [12] and *query generation* [29]. The pairwise approach prompts LLM with a pair of passages to a given query to indicate which is more relevant, using aggregation methods [24] or sorting algorithms [27, 36, 38] to derive the final ranking. The listwise approach aims to receive a query along with a list of candidates and directly generate a ranking list based on their relevance to the query [14, 30]. Recently, some studies have attempted to distill smaller listwise reranking models from existing powerful rerankers like RankGPT [25, 26, 37]. Our proposed method aims to enhance the efficiency of listwise approaches while preserving their effectiveness.

### 2.2 Context Compression

Context compression, which seeks to reduce the input length of LLMs while retaining the essential information from the original context, has recently garnered considerable attention. One approach is to heuristic modify the context to make it concise while retaining key information. LLMLingua [10] introduces a coarse-to-fine prompt compression method based on the perplexity score. RE-COMP [35] proposes compressing documents into text summaries for RAG. Another direction is to compress the text into dense slots or soft prompts, such as AutoCompressor [3], ICAE [6], and Gist [18]. However, these methods only compress a single prompt and are inadequate for ranking tasks. In contrast, our proposed method is specifically designed for ranking tasks and can be regarded as a variant of the second kind of method.

Recently, a contemporary work, xRAG, proposed using embedding models to compress a document into a token for RAG, which is similar to our proposed method [2]. Compared to it, our proposed PE-Rank method has the following differences: firstly, we compress prompts for the ranking task which is more complex, and secondly, we compress multiple documents as input at once.

## 3 METHODOLOGY

### 3.1 Overview

The overview architecture of PE-Rank is shown in Figure 2, we introduce the model under the two-stage ranking paradigm.

Specifically, we first use the dense retrieval model to pre-encode the corpus into a vector index. Given a query $q$, we use the same encoder to encode it into an embedding and retrieve several most relevant candidate passages $\mathcal{P}_{cand} = [p_1, ..., p_n]$ and their embeddings $e_{p_1}, ..., e_{p_n}$. Depending on the retrieval model, the embedding of [CLS] token or the mean pooling is used. Then vector similarity is used as the relevance score between query and passages.

In the reranking stage, our key idea is to take the embeddings from the previous stage as a good context compression of passages. Therefore, we propose replacing the original passage with

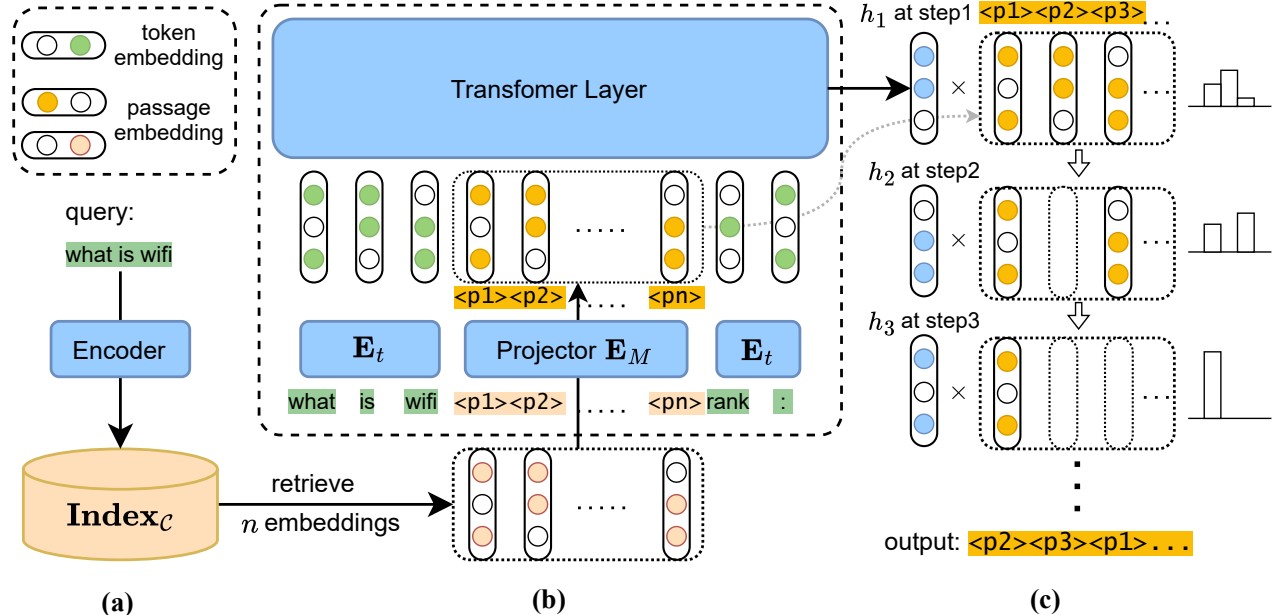

**Figure 2: Overview of PE-Rank under a two-stage ranking paradigm. (a) is retrieval stage, retrieve $n$ passage embeddings; (b) is the forward pass procedure of LLM; (c) shows the listwise decoding process.**

the single embedding representation as the input of LLMs. However, there are dimensional and distribution differences between the passage embeddings and LLM's token embeddings, which require us to bridge the gap between two spaces with a learned mapping function. Taking inspiration from previous work on aligning two modalities [13], we introduce a two-layer multi-layer perception (MLP), denoted as $\mathbf{E}_M$, as the mapping function. Here we treat these transformed embeddings $\mathbf{E}_M(\boldsymbol{e}_{p_i})$ as the embeddings of additional out-of-vocabulary special tokens, where one passage is represented as one special token, for example <p1> represents $p_1$.

Furthermore, by taking the instruction $I$ and query $q$ as normal tokens and then concatenating the token embeddings and transformed passage embeddings, we can define the simplified input embeddings of LLM at the first generation step:

$$\mathbf{E}_{\text{In}}^{(1)} = \mathbf{E}_t(I \oplus q) \oplus \mathbf{E}_M(\boldsymbol{e}_{p_1}) \cdots \oplus \mathbf{E}_M(\boldsymbol{e}_{p_n}), \quad (1)$$

where $\mathbf{E}_t$ is the token embedding layer of LLM. The complete prompts are listed in Appendix A. In the next section, we will introduce how to output the ranking list in detail.

It should be pointed out that although we describe PE-Rank in the background of two-stage ranking, it can be applied separately for reranking, simply using the encoder as a text compressor by encoding passages on the fly.

## 3.2 Inference

During inference, listwise rerankers aim to output a ranking list directly. For LLM-based listwise approaches, we usually generate the ranking list autoregressively. In previous work, LLMs are prompted to generate a string that could be parsed into a ranking list, such as

---

**Algorithm 1:** Dynamic-Constrained Decoding

**Input** : Candidates $\mathcal{P}_{cand} = [p_1, ..., p_n]$, Initial Input Embeddings $\mathbf{E}_{\text{In}}^{(1)}$

**Output**: Ranking List $\hat{\mathcal{P}}_{rank} = [\hat{p}_1, ..., \hat{p}_n]$

1  $\hat{\mathcal{P}}_{rank} \leftarrow \emptyset$
2  **for** $i \leftarrow 1$ **to** $n$ **do**
3  $\quad$ $\boldsymbol{h}_i \leftarrow \text{Transformer}(\mathbf{E}_{\text{In}}^{(i)})$
4  $\quad$ $\hat{p}_i \leftarrow \arg\max_{p \in \mathcal{P}_{cand}}(\boldsymbol{h}_i^T \cdot \mathbf{E}_M(\boldsymbol{e}_p))$
5  $\quad$ $\mathbf{E}_{\text{In}}^{(i+1)} \leftarrow \mathbf{E}_{\text{In}}^{(i)} \oplus \mathbf{E}_M(\boldsymbol{e}_{\hat{p}_i})$
6  $\quad$ $\mathcal{P}_{cand}.\text{remove}(\hat{p}_i)$
7  $\quad$ $\hat{\mathcal{P}}_{rank}.\text{append}(\hat{p}_i)$
8  **end**
9  **return** $\hat{\mathcal{P}}_{rank}$

---

"[2] > [3] > [1]..." [25, 30]. However, in early experiments, we found that generating a string representing ranking may be difficult and slow, as LLM may output in the wrong format or output useless content, such as explanation.

To address this issue, we propose a "Dynamic-Constrained Decoding" strategy in Algorithm 1. During decoding, we replace the original output layer over the whole vocabulary with the concatenation of embeddings of passages that need to be ranked, treating the embedding representation of those passages as a set of special tokens. Moreover, the decoding space, i.e., the output layer, is dynamically changed as the ranking process progresses, as fewer

remaining passages need to be ranked, resulting in a continuous decrease in decoding space.

At each generation step, we no longer output a normal numerical token but instead constrain the decoding space only in these special tokens, to perform accurate ranking. Therefore, we can directly output a list of tokens that represent the ranking of passages, such as "<p2><p3><p1>...". Furthermore, as the decoding space and the number of generated tokens are much smaller than the original vocabulary space, inference will be accelerated.

For example, as shown in Figure 2 (c), we first obtain the hidden state $\boldsymbol{h}_1$ from LLM in the first decoding step and calculate the output probability distribution with all the passages embeddings $\mathrm{E}_M(\boldsymbol{e}_{p_1}), ..., \mathrm{E}_M(\boldsymbol{e}_{p_n})$, then take the $p_2$ with the highest probability as the top-1 passage in the result. In the second decoding step, we append $\mathrm{E}_M(\boldsymbol{e}_{p_2})$ to the input embeddings of LLM at last, remove it from the decoding space, and use the hidden state $\boldsymbol{h}_2$ in the second step to get the next output. By repeating this process, we obtain the final ranking.

It's worth noting that there are existing works of constrained decoding [32], however, notable distinctions exist between our approach and theirs. Firstly, The decoding space of these related works is the original decoding space of LLM and is static, while that of our proposed method is outside the original vocabulary and dynamic. Secondly, These related works employed constrained decoding for generating text with strict format constraints. In contrast, our goal is simply to output a ranking list of tokens which leads to a more simple and more efficient method.

We use the greedy search algorithm in the actual inference process. It should be pointed out that when generating the next special token, the model relies on the previously predicted results rather than the ground truth.

## 3.3 Training

During training, we aim to address two challenges: aligning disparate embedding spaces and adapting the model for ranking. Consequently, we divide the training into two stages: (1) the alignment stage, which aligns the output space of the dense retrieval model with the token embedding space of the LLM, and (2) the learning-to-rank stage, which enables the model to acquire knowledge about ranking.

*Alignment stage.* At this stage, our objective is to ensure that the passage embeddings produced by the dense retrieval model are comprehensible to the large language model and effectively represent the original text information. To achieve this, we design a text reconstruction task for training. Given a piece of text $t$, it is first encoded into an embedding and passed through the MLP. Taking the transformed embedding as part of the input, the LLM is prompted to reconstruct the original text based on the embedding. The simplified input of LLM can be formalized as:

$$\mathrm{E}_{\text{In-Align}} = \mathrm{E}_t(I) \oplus \mathrm{E}_M(\boldsymbol{e}_t), \tag{2}$$

We employ language modeling loss for training:

$$\mathcal{L}_{\text{Align}} = -\sum_{i=1} \log P_\theta(t_i | \mathrm{E}_{\text{In-Align}} \oplus \mathrm{E}_t(t_{<i})). \tag{3}$$

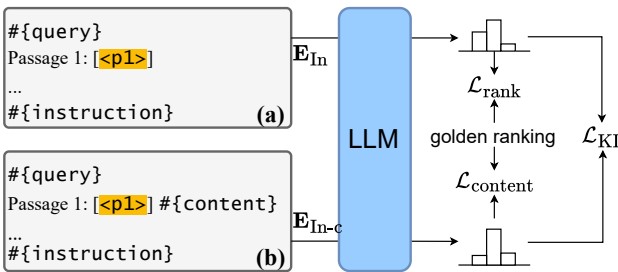

**Figure 3: Illustration of two types of training data and the learning-to-rank training process.**

Note that we freeze the encoder and the LLM and only fine-tune the parameters of MLP, that is, we only learn the mapping between two different embedding spaces, without changing themselves.

*Learning-to-rank stage.* Considering the decoding process, it can be viewed as a sequential ranking learning process: at each step, we provide the previously decoded rankings and maximize the probability of generating the next most relevant passage. Formally, if given a query $q$ and the golden ranking list $[p_1, ..., p_n]$, at step $i$, we maximize the conditional probability of $p_i$ given $q$ and $p_{<i}$:

$$
\begin{aligned}
P_\theta(p_i | q, p_{<i}) &= P_\theta(p_i | \mathrm{E}_{\text{In}}^{(i)}) \\
&= \frac{\exp(\boldsymbol{h}_i^T \cdot \mathrm{E}_M(\boldsymbol{e}_{p_i}))}{\sum_{j=i}^n \exp(\boldsymbol{h}_i^T \cdot \mathrm{E}_M(\boldsymbol{e}_{p_j}))},
\end{aligned}
\tag{4}
$$

where $\theta$ is the model's parameters. Considering the whole sequential process, it is equivalent to listwise learning to rank loss ListMLE [33]:

$$\mathcal{L}_{\text{rank}} = -\sum_{i=1}^n \log P_\theta(p_i | \mathrm{E}_{\text{In}}^{(i)}). \tag{5}$$

Here we only leverage the passage embeddings for ranking, as illustrated in the prompt (a) in Figure 3. The full prompts can be found in Appendix A.

However, understanding entire passages with single embedding and utilizing them for ranking may be challenging for LLMs, which may result in difficulties when directly training with Equation (5). Therefore, we incorporate both the original text and the passage embedding into the model inputs and apply the same forward pass to compute the loss:

$$\mathcal{L}_{\text{content}} = -\sum_{i=1}^n \log P_\theta(p_i | \mathrm{E}_{\text{In-c}}^{(i)}), \tag{6}$$

where $\mathrm{E}_{\text{In-c}}^{(i)}$ is defined similarly as Equation (1), but includes the content as part of the input, as illustrated in the prompt (b) in Figure 3. We believe this approach enhances the model's ability to utilize the token-level interactions between query and passage and helps transfer this ability when solely using embeddings for ranking.

Additionally, we also employ KL Divergence for distillation, which enables the model using compressed embeddings to emulate

the proficiency in handling the uncompressed texts:

$$\mathcal{L}_{\text{KL}} = \sum_{i=1}^{n} D_{\text{KL}}(P_\theta(p_i|\mathbf{E}_{\text{In}}^{(i)}) \| P_\theta(p_i|\mathbf{E}_{\text{In-c}}^{(i)})). \qquad (7)$$

The final loss function is defined as:

$$\mathcal{L} = \mathcal{L}_{\text{rank}} + \mathcal{L}_{\text{content}} + \alpha\mathcal{L}_{\text{KL}}. \qquad (8)$$

Here $\alpha$ is the hyperparameter. We fine-tune both MLP and LLM in this stage but keep the encoder frozen.

It is important to note that during training, we use the golden ranking labels at each step, which differs from the inference process.

## 4 EXPERIMENT SETUP

### 4.1 Model selection

We choose Mistral-7B-Instruct-v0.2 [9] as our backbone model since it has a strong instruction-following ability. For most experiments, we select one popular embedding model, i.e., Jina-Embeddings [7, 16], which has 137M parameters and shows a strong generalization ability across different corpora. Also, we use different embedding models in the ablation study to demonstrate that our framework can adapt to other models. We will use PE-Rank$_\star$ to denote different embedding models, but for convenience, if not indicated, Jina-Embeddings is used.

### 4.2 Training Data

During the alignment stage, we employ segmented Wikipedia as the training dataset. The texts in the Wikipedia dataset, authored and reviewed by humans, are of higher quality and completeness. Additionally, its comprehensive nature provides knowledge from diverse fields, rendering it reliable for training in the alignment stage. Specifically, we utilized the Wikipedia dump from Dec 2020, preprocessed by Izacard et al. [8], totaling around 31.5 million texts. We sampled 2 million data pieces for training. The complete data format can be found in Appendix A.

In the learning-to-rank stage, we utilize the MS MARCO dataset [1]. MS MARCO is a large-scale passage retrieval dataset that contains around 8.8 million passages and 800,000 queries, of which about 500,000 have manually annotated relevance labels.

We use Jina-embeddings-v2-base-en[1] as the retrieval model to retrieve the top 20 candidate passages for all queries in the training set, to construct the dataset. However, it only includes binary annotations (i.e., relevant or irrelevant) and cannot be directly used as training data for our training procedure. Therefore, following the approach of Zhang et al. [37], we use an existing powerful supervised reranking model, i.e., MiniLM[2] trained on MS MARCO, as the annotation model to approximate the golden ranking. Following Pradeep et al. [25], we used a data augmentation strategy of randomly shuffling document order.

To facilitate training, we excluded samples with excessively long lengths, retaining only those with input lengths less than 2048. Consequently, our dataset for this stage comprises 232,419 samples and each sample contains 20 passages and the approximated golden ranking.

---
[1]https://huggingface.co/jinaai/jina-embeddings-v2-base-en
[2]https://huggingface.co/cross-encoder/ms-marco-MiniLM-L-6-v2

### 4.3 Evaluation Datasets

We evaluate PE-Rank on multiple retrieval benchmarks, including TREC DL [4] and BEIR [31]. TREC DL uses the MS MARCO dataset [1] as the retrieval corpus and has fine-grained relevance annotations. We use the test sets of TREC DL 2019 and TREC DL 2020, which contain 43 and 54 queries respectively. BEIR contains 18 datasets from different fields with different query requirements, aiming to evaluate the generalization ability of ranking models. Following previous work [30], we conduct evaluations on 8 datasets that contain a relatively small number of queries. We use NDCG@10 as the evaluation metric.

### 4.4 Baselines

We select several existing methods as our basic baselines.

*Supervised Neural Rerankers.* First, we select two typical supervised models, including **monoBERT** [20] and **monoT5** [21]. Both of these models are trained on the MS MARCO dataset using a large number of human annotation labels.

*LLM-based Rerankers.* Additionally, we use one unsupervised LLM-based method as baselines:

- **RankGPT** [30], a state-of-the-art method that uses a sliding window strategy for listwise ranking based on GPT.

We also add listwise reranking models that are based on smaller LLMs (such as an LLM with 7B parameters) and are distilled from existing rerankers as baselines, including:

- **RankVicuna** [25], which is a listwise model based on Vicuna-7B, using RankGPT$_{3.5}$ as the teacher model.
- **RankZephyr** [26], which is a listwise model based on Zephyr-7B, using both RankGPT$_{3.5}$ and RankGPT$_4$ as the teacher model thus achieve a strong ranking performance.

However, a direct comparison with the above baseline is not intuitive because of the impact of different foundation models and training data. Furthermore, the underlying LLM and the training data are orthogonal to the ranking paradigm. Therefore, to ensure a fair comparison between the previous listwise ranking paradigm (e.g., RankGPT and RankVicuna) and PE-Rank, we retrained a listwise ranking model using the similar training process and paradigm as RankVicuna but based on the same LLM, i.e., Mistral-7B, and training data with PE-Rank, denoted as **RankMistral**, as a main baseline. We believe that directly comparing PE-Rank with RankMistral can provide more rich insights.

Also, we use this model to evaluate different text compression strategies and compare them with PE-Rank. Specifically, we can use different texts to replace the original passage in the inputs, denoted as **RankMistral$_*$**, where $*$ can be passage ($p$), summary ($s$), or title ($t$). These baselines help us evaluate the effectiveness and efficiency of different compression methods under a consistent setting.

### 4.5 Implementation Details

We implement all training codes based on the PyTorch framework. To optimize memory usage and accelerate training, we applied Deepspeed ZeRO stage 2 [28] and BFloat16 mixed precision techniques. Additionally, Flash attention [5] was used to further improve training efficiency. In the alignment stage, we trained the 7B Mistral

**Table 1: Results (NDCG@10) of reranking top-100 passages on BEIR benchmark. *Ret* means the retrieval model used in the first stage. In each block, i.e., when using the same retriever, * denotes that there is no statistically significant difference between PE-Rank and the baselines ($p \geq 0.05$ level) using a two-sided t-test. The best model among all is in bold, while the best model in each block is underlined.**

| Model | *Ret.* | Covid | NFCorpus | Touché | DBPedia | SciFact | Signal | News | Robust | Avg. |
|---|---|---|---|---|---|---|---|---|---|---|
| BM25 | - | 0.5947 | 0.3375 | 0.4422 | 0.3180 | 0.6789 | 0.3305 | 0.3952 | 0.4070 | 0.4380 |
| Jina-Embeddings | - | 0.6894 | 0.3143 | 0.2868 | 0.3332 | 0.6553 | 0.2576 | 0.3980 | 0.3823 | 0.4146 |
| monoBERT | | 0.7001 | 0.3688 | 0.3175 | 0.4187 | 0.7136 | 0.3144 | 0.4462 | 0.4935 | 0.4716 |
| monoT5 | BM25 | 0.8071 | **0.3897** | 0.3241 | 0.4445 | **0.7657** | 0.3255 | 0.4849 | 0.5671 | 0.5136 |
| RankGPT$_{3.5}$ | | 0.7667 | 0.3562 | 0.3618 | 0.4447 | 0.7043 | 0.3212 | 0.4885 | 0.5062 | 0.4937 |
| RankGPT$_4$ | | **0.8551** | 0.3847 | **0.3857** | **0.4712** | 0.7495 | **0.3440** | **0.5289** | **0.5755** | **0.5368** |
| RankMistral | BM25 | 0.7800* | 0.3310* | 0.2746* | 0.3771* | 0.6622* | 0.3004* | 0.3710 | 0.3954 | 0.4365 |
| PE-Rank | | 0.7772 | 0.3639 | 0.3306 | 0.4005 | 0.6938 | 0.3374 | 0.4970 | 0.4740 | 0.4843 |
| RankMistral | Jina | 0.8019* | 0.2974* | 0.2916* | 0.4025 | 0.6385* | 0.2817 | 0.3580 | 0.3569 | 0.4286 |
| PE-Rank | | 0.7749 | 0.3092 | 0.3000 | 0.3626 | 0.6448 | 0.2654 | 0.4478 | 0.4373 | 0.4428 |

**Table 2: Results (NDCG@10) of reranking top-100 passages on TREC DL. *Ret* means the retrieval model used in the first stage. In each block, * denotes that there is no statistically significant difference between PE-Rank and the baselines ($p \geq 0.05$ level) using a two-sided t-test.**

| Model | *Ret.* | TREC DL19 | TREC DL20 |
|---|---|---|---|
| BM25 | - | 0.5058 | 0.4796 |
| Jina-Embedding | - | 0.6594 | 0.6389 |
| *Supervised models trained with human annotation* | | | |
| monoBERT | BM25 | 0.7050 | 0.6728 |
| monoT5 | | 0.7183 | 0.6889 |
| *Unsupervised LLM-based listwise models* | | | |
| RankGPT$_{3.5}$ | BM25 | 0.6580 | 0.6291 |
| RankGPT$_4$ | | **0.7559** | **0.7056** |
| *LLM-based listwise models trained with distillation* | | | |
| RankVicuna | | 0.6682* | 0.6549* |
| RankZephyr | BM25 | 0.7420 | 0.7086 |
| RankMistral | | 0.7173* | 0.6807* |
| PE-Rank | | 0.7048 | 0.6354 |
| RankVicuna | | 0.6981* | 0.7061* |
| RankZephyr | Jina | 0.6983* | 0.7515 |
| RankMistral | | 0.7144* | 0.7327* |
| PE-Rank | | 0.7091 | 0.6948 |

model for 1 epoch with an effective batch size of 128 and a learning rate of $1 \times 10^{-4}$. In the learning-to-rank stage, we trained the model for 1 epoch with an effective batch size of 32 and a learning rate of $2 \times 10^{-5}$. $\alpha$ in Eqution (8) is set to 0.2 based on prior experiments. All models were trained on 4 Nvidia H100 GPUs. It is important to note that the hyperparameters were determined based on empirical observations, as comprehensive hyperparameter tuning was beyond the scope of this study due to resource constraints.

During the evaluation, for each dataset, we first use a retrieval model to recall the top 100 passages for each query, and then evaluate the reranking results. For convenience, we encode the passages on the fly, allowing us to use different retrieval models to provide a more comprehensive comparison. If not otherwise specified, we use the sliding window trick [30] to complete the whole ranking and set the window size to 20 and the step size to 10, therefore need 9 passes in total. We use one Nvidia H100 GPU to finish all evaluations.

## 5 EXPERIMENT RESULTS

### 5.1 Effectiveness Analysis

We first evaluate the effectiveness of PE-Rank on TREC DL and BEIR benchmarks, and present the results in Table 2 and Table 1. From the results, we can observe that the supervised models based on BERT and T5 can achieve competitive ranking performance, while in the LLM-based baselines, using the strongest LLM, GPT-4, for listwise reranking can achieve state-of-the-art across all models on three datasets. As for distilled models, RankZephyr also shows promising ranking effectiveness, and we attribute this to using GPT-4 as the teacher model.

Comparing the proposed PE-Rank model with other baselines, we can see that: (i) PE-Rank can approach supervised baselines' performance. (ii) Despite compressing the entire passage into a single embedding, PE-Rank maintains comparable results to the uncompressed distilled listwise models, especially RankMistral. Specifically, we can find that the ranking performance of PE-Rank on both DL19 and DL20 has no statistically significant difference compared with RankMistral. On BEIR, there is also no significant difference on most datasets, even on some datasets PE-Rank surpassing RankMistral. This observation indicates that under the same settings, i.e., the same LLM and training data, PE-Rank can achieve comparable effectiveness to the previous listwise ranking paradigm.

It should be emphasized that when PE-Rank remains competitive, it has a significant efficiency advantage, and we will provide a detailed analysis in the next section.

**Table 3: Efficiency analysis for reranking top $n$ candidates retrieved by BM25 on TREC DL19 and Covid. # Proc and # Gen mean the number of processed tokens in the prefill stage and generated tokens in the decode stage, respectively. For PE-Rank, we also include the time for encoding the passages on the fly. $L_p$ of DL19 and Covid is approximately 100 and 423. In each block, \* denotes no statistically significant difference between the compression settings and RankMistral$_p$ ($p \geq 0.05$ level) using a two-sided t-test. The subscript of RankMistral means the form of inputs, including original passage ($p$), summary ($s$), or title ($t$).**

| Model | $n$ | TREC DL19 | | | | Covid | | | |
|---|---|---|---|---|---|---|---|---|---|
| | | NDCG@10 | # Proc. | # Gen. | Latency (s) | NDCG@10 | # Proc. | # Gen. | Latency (s) |
| RankMistral$_p$ | | **0.6465** | 2265.8 | 109.9 | 2.04 | 0.7090 | 8190.9 | 110.4 | 2.51 |
| RankMistral$_s$ | 20 | 0.6303* | 1490.7 | 106.1 | 1.99 (×.98) | 0.6515 | 2224.2 | 100.2 | 1.92 (×.76) |
| RankMistral$_t$ | | 0.4862 | 409.5 | 107.2 | 1.93 (×.95) | 0.6671* | 829.7 | 110.4 | 1.89 (×.75) |
| PE-Rank | | 0.6266* | 326.9 | 20.0 | 0.42 (×.21) | **0.7234*** | 344.3 | 20.0 | 0.44 (×.18) |
| RankMistral$_p$ | | **0.7196** | 19506.2 | 910.2 | 16.20 | **0.7780** | 71431.2 | 986.5 | 21.46 |
| RankMistral$_s$ | 100 | 0.7050* | 13485.3 | 881.6 | 15.68 (×.97) | 0.7385* | 20148.6 | 929.6 | 16.94 (×.79) |
| RankMistral$_t$ | | 0.4543 | 3753.4 | 865.1 | 15.12 (×.93) | 0.7540* | 7555.0 | 916.9 | 15.87 (×.74) |
| PE-Rank | | 0.7048* | 2942.4 | 180.0 | 3.62 (×.22) | 0.7772* | 3098.9 | 180.0 | 3.65 (×.17) |

## 5.2 Efficiency Analysis

We conduct efficiency analysis from the perspectives of consumed tokens and latency. Here we conduct experiments on the TREC DL19 dataset and one of the datasets of BEIR, i.e., Covid. TREC DL19 and DL20 have the same corpus and similar distribution, therefore we only show the results on one. Instead, we select the Covid dataset as an alternative since it has longer documents.

*Number of Consumed Tokens.* We theoretically analyze the number of *processed* tokens in the prefill stage and *generated* tokens in the decode stage of different methods. Assume a single pass with $n$ passages of average length $L_p$ and instruction of length $L_I$, methods based on the text like RankGPT exhibit an input length of $O(L_I + nL_p)$, which increases almost proportionally with $L_p$. In contrast, PE-Rank shows an input length of $O(L_I + n)$ which will be unchanged when $L_p$ increases. For RankGPT-like methods, they need to generate numbers as well as identifiers such as "[]" and may not output completely correctly, resulting in the number of generated tokens for $\Omega(mn)$. In practice $m \approx 4.5$. As for PE-Rank, by employing the DC decoding method, the number is exactly equal to $n$ since only $n$ unique special tokens will be output.

It is important to note that when employing the sliding window strategy, the above results must be multiplied by the times of sliding. However, PE-Rank, due to the compression of input length, can achieve completion with fewer times or even in a single pass, thereby further underscoring its efficiency advantages.

Table 3 displays the number of tokens consumed by different methods. The results show that, although simple text compression techniques partially reduce tokens to be processed, they may lead to performance degradation. Specifically, when using titles as compression on DL19, i.e., RankMistral$_t$, the performance is even lower than BM25, possibly due to title misses or lack of valid information. Using summaries as input also results in performance loss, particularly on the Covid dataset. Besides, these text-based methods do not decrease the number of generated tokens. Note that the model may not output in the required format in practice, leading to fluctuations in the number of generated tokens. In contrast, PE-Rank significantly reduces the number of tokens to be processed and generated,

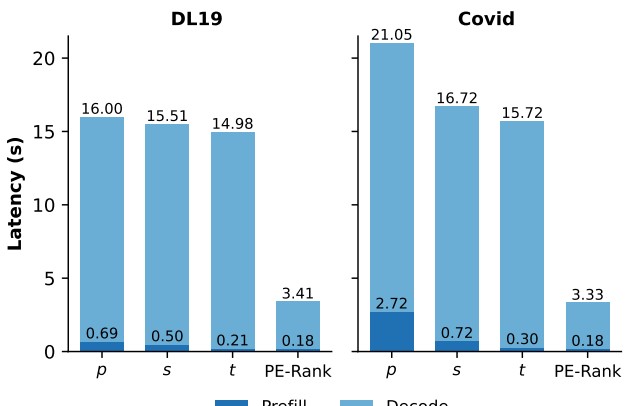

**Figure 4: Latency of reranking top 100 candidates at different stages during inference. $p$ (passage), $s$ (summary), and $t$ (title) denote the different forms of inputs of RankMistral.**

while there is no statistically significant difference compared with RankMistral$_p$ on all datasets and reranking settings.

*Latency.* We also analyze the reranking latency using different methods in Table 3. The results indicate that heuristic text compression techniques, such as using titles or summaries, do not significantly reduce latency. Conversely, by leveraging passage embedding as a text compression representation, PE-Rank markedly accelerates the ranking process, achieving approximately a five-fold increase in speed across different candidate numbers and datasets, with only about 0.2 times the delay of the uncompressed method. Notably, when reranking the top 20 candidates, the ranking latency for a single query can be reduced to 0.5 seconds, which for the first time makes the LLM-based listwise ranking method practical for being employed in an online search system.

To fully comprehend the efficiency advantages of PE-Rank, we subdivide the sources of latency into prefilling and decoding, and conduct a more detailed analysis, as shown in Figure 4. Our findings

**Table 4: Ablation on different training strategies. We show the results of ranking top 100 candidates of BM25.**

|  | DL19 | DL20 | Covid | News |
|---|---|---|---|---|
| (a) PE-Rank | **0.7048** | 0.6354 | **0.7772** | 0.4740 |
| (b) w/o Alignment | 0.6583 | 0.6135 | 0.7312 | 0.4671 |
| (c) w/o $\mathcal{L}_{content}$ & $\mathcal{L}_{KL}$ | 0.6843 | **0.6442** | 0.7721 | 0.4623 |
| (d) w/o $\mathcal{L}_{KL}$ | 0.6843 | 0.6403 | 0.7633 | **0.4742** |
| (e) w/o $\mathcal{L}_{content}$ | 0.6666 | 0.6085 | 0.7594 | 0.4715 |

first indicate that latency predominantly arises from decoding, with prefilling contributing only minimally. On datasets with shorter passage lengths, such as DL19, PE-Rank does not demonstrate a significant efficiency advantage during the prefilling stage; instead, the advantage is primarily observed in decoding, as fewer tokens need to be output, as previously analyzed. As passage length increases, given that the input length for PE-Rank does not increase linearly, it also exhibits efficiency advantages in prefilling, as the results observed on Covid.

## 5.3 Ablation Study

*5.3.1 Training Strategies.* We analyze the impact of various training strategies on PE-Rank's ranking performance, with results presented in Table 4. As expected, the model encompassing all training stages and loss functions exhibited the highest performance across four datasets. Additionally, we make the following observations: firstly, the alignment stage markedly influences ranking performance, though a model with ranking capabilities can still be obtained without it. Secondly, adding text without the KL loss (row (d) vs. (c)) or merely incorporating the KL loss (row (e) vs. (c)) during training does not yield substantial improvements. Consequently, we infer that it is imperative for PE-Rank to comprehend the token-level interaction between query and passages, as well as to simulate the original text only using passage embeddings.

*5.3.2 Different Embedding Models.* To verify whether our proposed framework can generalize to different embedding models, we choose a different embedding model for experiments. Specifically, we select BGE-base [34], a BERT-based model that achieves the top tier position across the same parameter scale models on the MTEB benchmark [19]. We use BGE as the embedding model and the same complete training process as Jina-Embeddings to obtain a new model. The results are shown in Table 5.

Firstly, using Jina-Embeddings and BGE as the encoder and leveraging their passage embeddings for reranking are both effective, reranking the candidates obtained from different retrieval models on different datasets can consistently bring improvement. This demonstrates that the PE-Rank approach can be applied to different embedding models.

However, although BGE scores higher than Jina-embedding on MTEB, the performance of reranking BM25 retrieval results using BGE embeddings is consistently lower across three different datasets compared to using Jina embeddings. Due to the use of different training data and pooling methods in these two models, it is challenging to directly determine the cause of this discrepancy.

**Table 5: Using different embedding models to obtain passage embeddings as context compression.**

| Model | Ret. | DL19 | DL20 | BEIR Avg. |
|---|---|---|---|---|
| BM25 |  | 0.5058 | 0.4796 | 0.4380 |
| PE-Rank$_{Jina}$ | BM25 | 0.7048 | 0.6354 | 0.4843 |
| PE-Rank$_{BGE}$ |  | 0.6728 | 0.6352 | 0.4791 |
| Jina-Embeddings | Jina | 0.6594 | 0.6389 | 0.4146 |
| PE-Rank$_{Jina}$ |  | 0.7091 | 0.6948 | 0.4428 |
| BGE-base | BGE | 0.7022 | 0.6621 | 0.4514 |
| PE-Rank$_{BGE}$ |  | 0.7293 | 0.6780 | 0.4600 |

**Table 6: The impact of different settings in the sliding window strategy on effectiveness and efficiency of reranking top 100 candidates retrieved by BM25.**

| Model | NDCG | $w/s$ | #Proc. | Latency |
|---|---|---|---|---|
| RankMistral$_p$ | 0.7196 | 20 / 10 | 19510.2 | 16.72 |
|  | 0.6026 | 40 / 20 | 17152.3 | 9.10 |
|  | 0.5154 | 100 / - | 10561.9 | 4.09 |
| PE-Rank | 0.7048 | 20 / 10 | 2942.4 | 3.68 |
|  | 0.7012 | 40 / 20 | 2187.7 | 3.05 |
|  | 0.6857 | 100 / - | 1210.9 | 1.90 |

Nonetheless, we have reason to believe that models excelling in general embedding benchmarks may not necessarily perform well in this context. This issue is worth further investigation.

*5.3.3 Impact of Sliding Window.* We investigate the effects of varying window sizes ($w$) and step sizes ($s$) in sliding window strategies, with results presented in Table 6. For RankMistral, ranking performance decreases sharply as window size increases. This is attributable to two factors: firstly, RankMistral struggles to manage long contexts containing rich information; secondly, it is trained on data with a window size of 20, which may prevent it from generating complete rankings with larger window sizes. In contrast, PE-Rank effectively addresses these issues. The compressed text maintains a shorter total length, and the compressed representation, i.e., passage embeddings, remains the key information of the original text. Additionally, the DC decoding method ensures accurate output of complete rankings. Consequently, PE-Rank's ranking performance remains relatively stable. More importantly, PE-Rank can reduce the number of sliding windows, thereby enhancing ranking efficiency.

## 6 CONCLUSION

In this paper, we propose a novel approach, PE-Rank, for efficient listwise passage reranking with large language models, leveraging passage embedding as the context compression, as well as effective inference and training methods. Experiment results demonstrate that PE-Rank offers notable efficiency advantages and is practical for being employed in real search systems while achieving competitive reranking effectiveness.

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

## A  PROMPTS

**User:**
I will provide you with {{n}} passages, each with a special token representing the passage enclosed in [], followed by the original text.

Rank the passages based on their relevance to the search query: {{query}}.

Passage 1: [{{embedding}}] {{content}}

...

Passage {{n}}: [{{embedding}}] {{content}}

Search Query: {{query}}

Rank the {{n}} passages above based on their relevance to the search query in descending order. Only output the {{n}} unique special token in the ranking.

**Table 9: Data format used for learning-to-rank stage training.**

**User:**
- Given the passage: {{embedding}}, reconstruct the original text.
- Passage: {{embedding}} means the same as
- Passage: {{embedding}} Can you say the above text again?
- {{embedding}} Please provide a reconstruction of the preceding passage.
- Passage: {{embedding}} is about what?
- {{embedding}} Could you give me a different version of the passage above?
- Passage: {{embedding}} Please offer a restatement of the provided passage.
- Passage: {{embedding}}, which means:

**Assistant:**
{{text}}

**Table 7: Prompts used for alignment stage training, where {{embedding}} and {{text}} are placeholders for transformed embeddings $E_M(e_t)$ and the original text $t$.**

**User:**

I will provide you with {{n}} passages. Rank the passages based on their relevance to the search query: {{query}}.
Passage 1: {{content}}

...

Passage {{n}}: [{{embedding}}] {{content}}

Search Query: {{query}}

Rank the {{n}} passages above based on their relevance to the search query in descending order. The output format should be [] > [] > ..., e.g., [4] > [2] > ..., Only respond with the ranking results with {{n}} unique numbers, do not say anything else or explain.

**Table 10: Data format used for training RankMistral.**

**User:**

I will provide you with {{n}} passages, each with a special token representing the passage enclosed in [].

Rank the passages based on their relevance to the search query: {{query}}.

Passage 1: [{{embedding}}]

...

Passage {{n}}: [{{embedding}}]

Search Query: {{query}}

Rank the {{n}} passages above based on their relevance to the search query in descending order. Only output the {{n}} unique special token in the ranking.

**Table 8: Data format used for learning-to-rank stage training.**

**User:**
Summarize the following passage, only output the summary, do not include anything else.
Passage: {{content}}

**Table 11: Prompts used for generating summary using Mistral-7B.**

