# OpenReview forum: "Leveraging Passage Embeddings for Efficient Listwise Reranking with Large Language Models"
_ACM.org/TheWebConf/2025/Conference — WWW 2025 Poster_

### Official Review · Reviewer_Z2Ty · 2024-11-17

**Novelty:** 5
**Technical Quality:** 5

**Review:**

The paper is well-written, clearly presented, and structured in a logical manner, making it accessible to a broad audience. Its originality lies in the integration of context compression techniques to mitigate input length constraints and the use of dynamic constrained decoding to improve output formatting.

### Pros:
1. The paper is well-written, easy to follow, and structured logically.
2. Integration of context compression techniques into ranking to overcome input length limitations is innovative.
3. Clever use of dynamic constrained decoding ensures the uniqueness of passages to be ranked and avoids the need for post-processing.
4. The latency reduction achieved by PE-Rank makes it a competitive choice for tasks requiring faster processing, even if retrieval effectiveness is slightly lower than state-of-the-art.


### Cons:
1. The methodology description is a little bit unclear to me, hoping for clarifying my confusion in below.
2. PE-Rank’s performance does not surpass models like RankGPT in terms of retrieval effectiveness. This could reduce its applicability in scenarios where retrieval accuracy is critical.

**Questions:**

I have a question on the methodology implementation and another about the model effectiveness.
1. Can you please explain more about the 'special token' in line 267? Do you refer to unspecified tokens in the vocabulary as 'out-of-vocabulary'? If so, would the limited number of special tokens affect the number of passages to be ranked and then further affect the retrieval performance? Alternatively, if you are extending the entire vocabulary by adding tokens, why is there no need to train embeddings for these new tokens?

2. Regarding Table 2, how does PE-rank compare to a lightweight retrieval system such as monoT5 + BM25? Could you clarify the advantages of using PE-rank in this context?

**Reviewer Confidence:**

3: The reviewer is confident but not certain that the evaluation is correct

**Scope:**

4: The work is relevant to the Web and to the track, and is of broad interest to the community

---

### Official Review · Reviewer_mVhQ · 2024-11-22

**Novelty:** 5
**Technical Quality:** 5

**Review:**

**Summary：**
This work introduces the PE-Rank framework to enhance the efficiency of listwise reranking while maintaining competitive ranking performance. Specifically, it utilizes passage embeddings as a context compression method. Additionally, the framework combines a two-stage training strategy with a novel decoding method to achieve efficient listwise passage reranking.

**Strengths:**
1. The authors conducted detailed experiments to analyze the efficiency of PE-Rank.
2. The idea of treating a passage as a special token is interesting and enables compressing multiple passages simultaneously for reranking.

**Weaknesses:**
1. This work is hard to follow due to poor readability. The inconsistency in symbols across figures, tables, and formulas, combined with incomplete descriptions, makes it confusing. For instance, in the second diagram of Figure 1, it is unclear whether the label "[<p1>]" represents a passage embedding or a token, as the authors do not provide a clear explanation. Additionally, Step 7 in Algorithm 1 is not clearly reflected in Figure 2. In Equation (1), the symbol "⊕" is not explicitly explained; while it may represent concatenation, the work should clearly define this symbol.
2. This work does not provide a reasonable explanation for why directly comparing PE-Rank with RankMistral would yield richer insights. The authors should clarify this point.
3. The "Dynamic-Constrained Decoding" strategy selects the passage from the candidate passages with the highest probability at each step as the selected one. While this strategy ensures local optimality, it cannot guarantee global optimality.

**Questions:**

1. Why are RankVicuna and RankZephyr included for comparison on TREC DL but not included for comparison on the BEIR benchmark?
2. In Figure 1, what does the orange label `<P1>` represent, and how does it differ from the label `<P1>` in Figure 3? Do they refer to the same concept (e.g., passage embedding or token), or do they have different meanings?
3. For Weakness 3, how does the strategy ensure that the final global reranking list achieves global optimality?
4. In Table 5, why do the model names in the 'Model' column include subscripts for embedding models, while the 'Ret.' column also lists embedding models? What do they specifically refer to?

**Reviewer Confidence:**

4: The reviewer is certain that the evaluation is correct and very familiar with the relevant literature

**Scope:**

4: The work is relevant to the Web and to the track, and is of broad interest to the community

---

### Official Review · Reviewer_o38Y · 2024-11-27

**Novelty:** 5
**Technical Quality:** 5

**Review:**

This paper introduces PERank, an efficient listwise passage reranking method. By using single passage embeddings as compressed context and treating each passage as a special token, PERank reduces input length and accelerates decoding through dynamic constraints. Experiments on multiple benchmarks show that PERank significantly improves efficiency in both prefilling and decoding while maintaining competitive ranking performance.

The motivation is clearly presented. But there are some confusing points in the experimental design. The author stated that there are drawbacks on RankGPT, but given the performance, especially in Tab 1, it is hard to say in what way PERank is better than RankGPT; or the author should find a way to fairly compare these two models. Even, RankGPT_{3.5} has a higher performance on average, and it is unsupervised with very low cost, while PERank, though the number of parameter is smaller, it needs training. It makes more sense to highlight that PERank is better among methods from the same category, such as RankVicuna, i.e., tab 2 and 3.

In general, this paper is comprehensive. Adding interesting case studies would make it better.

**Questions:**

What is the estimation of total training hours on 4 H100 cards?
Did the author compare with other zero-shot or unsupervised small LLMs with 7b parameters?
Did the author try PERank with larger LLMs as the backbone, would it be better than RankGPT?

**Reviewer Confidence:**

3: The reviewer is confident but not certain that the evaluation is correct

**Scope:**

4: The work is relevant to the Web and to the track, and is of broad interest to the community

---

### Official Review · Reviewer_WrjU · 2024-12-02

**Novelty:** 5
**Technical Quality:** 5

**Review:**

The paper introduces PE-Rank, a novel method for listwise passage reranking with LLMs that leverages passage embeddings as a context compression mechanism. It addresses the efficiency limitations of traditional approaches like RankGPT by treating passage embeddings as special tokens within Large Language Models. It contains two major key contributions, i.e., dynamic-constrained decoding and two-Stage training. Expermental results demonstrate the efficiency of the proposed methods compared to the existing methods.

Pros:
1. This paper studies the problem of listwise passage reranking with LLMs, which could be a promising direction in the information retrieval domain.
2. The paper proposes PE-Rank that focuses on the context compression for efficiency optimization, which is a practical consideration.
3. The paper is generally written with fair clarity and thus is easy to follow.


Cons:
1. The primary concern with this work lies in its underwhelming performance compared to the unsupervised model, such as RankGPT3.5. The (exact) model sizes of RankGPT3.5 and PE-Rank remain unclear but are crucial to assess the potential justification for this performance gap.
2. The annotations in Tables 1 and 2 are somewhat confusing, particularly the statement that "* denotes that there is no statistically significant difference...". For certain dataset metrics, such as the results on Touche (e.g., 0.2746 vs. 0.3306), the numerical discrepancy appears substantial, making it challenging to reject the hypothesis of a significant difference (with the empirical-based guess). It is recommended that the authors clarify this discrepancy and provide additional context for these annotations.
3. It is unclear how PE-Rank's performance in terms of effectiveness and efficiency would scale when the number of processed and generated tokens in the prefill and decode stages expands. A discussion addressing this aspect would be beneficial to understand the model's scalability.

**Questions:**

Please see the weakness for details.

**Reviewer Confidence:**

3: The reviewer is confident but not certain that the evaluation is correct

**Scope:**

4: The work is relevant to the Web and to the track, and is of broad interest to the community